# Assessment of Antibiotic Resistance in Pediatric Infections: A Romanian Case Study on Pathogen Prevalence and Effective Treatments

**DOI:** 10.3390/antibiotics13090879

**Published:** 2024-09-13

**Authors:** Maria Madalina Singer, Renata Maria Văruț, Cristina Popescu, Kristina Radivojevic, Luciana Teodora Rotaru, Damian Roni Octavian, Banicioiu Mihai-Covei, Mihaela Popescu, Oancea Andreea Irina, Dragos Oancea, Alin Iulian Silviu Popescu, Cristina Elena Singer

**Affiliations:** 1Dermatology Department, Central Military Hospital, 010825 Bucharest, Romania; maria.singer1987@gmail.com; 2Research Methodology Department, Faculty of Pharmacy, University of Medicine and Pharmacy of Craiova, 200349 Craiova, Romania; 3Department of Anatomy, University of Medicine and Pharmacy, Discipline of Anatomy, 200349 Craiova, Romania; cristina.popescu@umfcv.ro; 4Faculty of Pharmacy, University of Medicine and Pharmacy of Craiova, 200349 Craiova, Romania; kristinaradivojevic03@gmail.com; 5Emergency Medicine and First Aid Department, Faculty of Medicine, University of Medicine and Pharmacy of Craiova, 200349 Craiova, Romania; luciana.rotaru@umfcv.ro (L.T.R.); ronidamian@yahoo.com (D.R.O.);; 6Department of Endocrinology, University of Medicine and Pharmacy of Craiova, 200349 Craiova, Romania; 7Department of Mother and Baby, Alessandrescu-Rusescu National Institute for Mother and Child Heatlth, 020395 Bucharest, Romania; oanceairina31@yahoo.com; 8Department of Gastroenterology, Fundeni Clinical Hospital, 022328 Bucharest, Romania; dragos.mihai02@yahoo.com; 9Department of Internal Medicine, University of Medicine and Pharmacy of Craiova, 200349 Craiova, Romania; alin.popescu@umfcv.ro; 10Department of Mother and Baby, University of Medicine and Pharmacy of Craiova, 200349 Craiova, Romania; cristina.singer@umfcv.ro

**Keywords:** antibiotic resistance, pediatric infections, bacterial pathogens, antibiogram

## Abstract

Antibiotic misuse in Romania has exacerbated the issue of antibiotic resistance, as patients often use antibiotics without proper medical consultation. This study aimed to assess the resistance of prevalent bacteria to different antibiotics. In this observational study conducted over six months, we analyzed 31 pediatric patients aged from 12 days to 13 years using the disk diffusion method. We identified 31 bacterial isolates, including 8 Gram-negative and 8 Gram-positive strains, with the most common being *Pseudomonas aeruginosa*, *Escherichia coli*, *Streptococcus pneumoniae*, methicillin-resistant *Staphylococcus aureus*, *Streptococcus* species, and *Elizabethkingia meningoseptica*. Our findings revealed that the most effective antibiotics were linezolid, ertapenem, and teicoplanin. In contrast, nearly all tested bacteria exhibited resistance to penicillin, followed by oxacillin and ampicillin. Resistance to cephalosporins varied with generation, showing higher resistance to lower-generation cephalosporins. The study highlights significant antibiotic resistance among common bacterial pathogens in Romanian pediatric patients, emphasizing the urgent need for controlled antibiotic use and alternative treatment strategies to combat this growing issue. Effective antibiotics such as linezolid and ertapenem offer potential solutions, whereas reliance on penicillin and lower-generation cephalosporins is increasingly futile.

## 1. Introduction

Antimicrobial resistance (AMR) and the continued presence of resistant strains increase the risk of treatment failure and recurrent infections [1]. Infections due to antibiotic-resistant bacteria are currently responsible for around 700,000 deaths worldwide each year. This figure is expected to exceed 10 million annual deaths by 2050 [2,3]. The availability of antibiotic treatments has significantly lowered mortality rates, leading to an overall rise in life expectancy [4]. However, the improper use of antibiotics has led to the emergence of multidrug-resistant (MDR) bacteria [5]. A strong association between AMR and various socioeconomic factors has been observed [6], highlighting a bidirectional relationship between antibiotic consumption and the development of resistance in both animals and humans, particularly in low- and low–middle-income countries (LMICs). This emphasizes the urgent need for integrated control strategies aimed at preventing transmission across multiple sectors within the One Health framework [7], and the necessity for enhanced surveillance and control measures to combat AMR in LMICs [8,9]. In clinical environments, conducting an antibiogram prior to prescribing antibiotics allows healthcare providers to choose the most appropriate antibiotic by identifying the specific bacteria responsible for the infection and their sensitivity to various antibiotics, thereby ensuring optimal treatment outcomes [10]. Additionally, tailoring treatments based on antibiogram results helps combat antibiotic resistance [11], reduces the likelihood of promoting antibiotic-resistant bacterial strains, decreases the incidence of related adverse reactions, and shortens hospital stays [12]. In Romania, the issue of AMR is critical, with 4300 deaths attributable to AMR and 16,500 deaths associated with it reported in 2019. This makes Romania the country with the highest age-standardized mortality rate due to AMR in the Central Europe GBD region, surpassing deaths from various other diseases including digestive and respiratory conditions. Notably, five key pathogens are particularly concerning due to their association with AMR deaths: *Escherichia coli* (*E. coli*) (4000 deaths), *Staphylococcus aureus* (*S. aureus*) (3100), *Klebsiella pneumoniae* (2300), *Pseudomonas aeruginosa* (*P. aeruginosa*) (1700), and *Streptococcus pneumoniae* (*S. pneumoniae*) (1200). These pathogens are known to cause severe infections such as bloodstream infections, peritoneal and intra-abdominal infections, lower respiratory infections, and thoracic-related infections [13]. Our study, which included performing antibiograms on 31 pediatric patients ranging in age from 12 days to 13 years, aimed to identify the most effective antibiotics against some of the most common bacteria and determine which are the most common bacterial strains. We detected a total of 16 different bacteria, most frequently including methicillin-resistant *Staphylococcus aureus* (MRSA) (11 cases), *P. aeruginosa* (10 cases), *Klebsiella* sp., (10 cases), *S. pneumoniae* (9 cases), and *E. coli* (7 cases). These findings closely align with national data, highlighting the predominance of these pathogens in AMR-related infections. In our study, more than 50 antibiotics were tested across major classes, including penicillins, cephalosporins, carbapenems, macrolides, tetracyclines, aminoglycosides, fluoroquinolones, sulfonamides, oxazolidinones, polypeptides, nitrofurans, and glycopeptides. Based on the data from all patients, the results revealed that the most effective antibiotics were linezolid, ertapenem, and teicoplanin. Conversely, most bacterial strains were resistant to penicillin, oxacillin, and ampicillin. Additionally, resistance to cephalosporins varied by generation, with higher resistance observed in lower-generation cephalosporins. This targeted approach to antibiotic selection is crucial in combating AMR. Understanding which bacteria are most common and their resistance patterns helps in crafting effective treatment protocols and informs public health strategies. Our study underscores the urgent need for continued surveillance and the strategic use of antibiotics to mitigate the rising threat of AMR, particularly in countries like Romania, where the impact is profoundly felt [14].

## 2. Results

This study aimed to determine bacterial resistance patterns in our patients, examining a diverse array of age groups and diagnoses. From infants merely days old to adolescents on the cusp of adulthood at 13 years, this study captured a wide range of cases. Throughout the study, a plethora of pathogens emerged (Table 1), including but not limited to *E. coli*, *Klebsiella*, MRSA, *Acinetobacter* spp., and *Proteus* spp. The primary diagnoses encompassed a myriad of ailments, such as bronchopneumonia, acute respiratory infections (ARI), sepsis, urinary tract infections (UTIs), and various other infections (Figure 1). The prevalence and resistance patterns of these pathogens underscore the complexity of infectious diseases in pediatric patients, necessitating a nuanced approach to treatment strategies. Through comprehensive analysis, this study aimed to elucidate the most effective antibiotics against these pathogens, thereby informing more targeted and efficacious treatment modalities for pediatric infections (Figure 2). Each diagnosis presented a unique challenge, with pathogens exhibiting varying degrees of resistance. Infants as young as 12 days old were afflicted with bronchopneumonia, ARI, and acute rhinopharyngitis, grappling with pathogens such as MSSA, *S. pneumoniae*, and *Streptococcus* sp. Meanwhile, toddlers at the age of one year faced acute viral pneumonia, ARI, and severe dystrophy, contending with formidable adversaries like *Acinetobacter baumannii* (*A. baumannii*), MRSA, *P. aeruginosa*, and *Elizabethkingia meningoseptica* (*E. meningoseptica*). The microbial landscape remained dynamic across different age groups and diagnoses. From UTIs and acute diarrheal illnesses in three-month-old infants to sepsis originating from the lungs and bronchopneumonia in four-year-olds, each case presented a unique pathogenic profile. Additionally, older children were not spared, with cases ranging from bullous impetigo and acute conjunctivitis to sepsis accompanying acute lymphoblastic leukemia (Figure 3A,B).

In the cases of MRSA, among the antibiotics showing consistently high activity (Figure 4) in 10 patients were tigecycline and linezolid. These antibiotics consistently demonstrated effectiveness against MRSA infections, indicating their potency in treating such cases. On the other hand, several antibiotics consistently showed resistance across all 10 patients, including oxacillin, penicillin, and rifampicin, indicating their limited efficacy against MRSA infections in this patient cohort. Overall, tigecycline and linezolid emerged as the most promising antibiotics for treating MRSA infections in this dataset, showcasing consistently high activity against the bacteria. Conversely, oxacillin, penicillin, and rifampicin appeared to be the least effective options, consistently demonstrating resistance across all patients (Table 2). This highlights the importance of judicious antibiotic selection based on susceptibility patterns to optimize treatment outcomes in MRSA infections.

The ten most active antibiotics were as follows: linezolid—10 patients; tigecycline—10 patients; gentamicin—9 patients; levofloxacin—8 patients; ofloxacin—8 patients; teicoplanin—7 patients; amikacin—7 patients; daptomycin—7 patients. Resistant antibiotics: oxacillin—10 patients; penicillin—10 patients; rifampicin—9 patients; Biseptol (trimethoprim/sulfamethoxazole)—8 patients; clarithromycin—8 patients; erythromycin—8 patients; clindamycin—8 patients; doxycycline—8 patients; tetracycline—8 patients; ampicillin—7 patients.

*P. aeruginosa* infections were assessed using a broad spectrum of antibiotics. The antibiotics tested included ciprofloxacin, chloramphenicol, meropenem, amikacin, aztreonam, colistin, gentamicin, imipenem, levofloxacin, ofloxacin, ticarcillin, tazocin (piperacillin/tazobactam), tobramycin, cefepime, ceftazidime, ceftazidime + avibactam, cefuroxime, clindamycin, doxycycline, erythromycin, Biseptol (trimethoprim/sulfamethoxazole), and rifampicin. This analysis highlights the importance of tailored antibiotic therapy based on individual susceptibility profiles. Meropenem, gentamicin, colistin, ciprofloxacin, levofloxacin, and tazocin were among the most effective antibiotics, demonstrating high activity across a significant number of patients. Conversely, cefepime, imipenem, and piperacillin/tazobactam showed varying levels of resistance, indicating the need for careful selection based on susceptibility testing to optimize treatment outcomes. Notably, some antibiotics exhibited both high activity in certain patients and resistance in others, underscoring the complexity of treating *P. aeruginosa* infections. For instance, ciprofloxacin demonstrated high activity in six patients (patients 1, 3, 16, 19, 21, and 26) but showed resistance in one patient (patient 14). Similarly, meropenem exhibited high activity in seven patients (patients 1, 3, 14, 21, 26, 30, and 31) while being resistant in one patient (patient 16). Imipenem showed high activity in five patients (patients 1, 3, 14, 16, 21, and 26) but was resistant in two patients (patients 13 and 31). This variability highlights the importance of conducting susceptibility testing for each patient to ensure the selection of the most effective antibiotic therapy. Resistance to carbapenems in this study is attributed to the presence of a type of *P. aeruginosa* that produces carbapenemases, and resistance was observed exclusively in these cases. Based on the data provided, the most efficient antibiotics for treating *P. aeruginosa* infections include meropenem, gentamicin, colistin, imipenem, amikacin, and tobramycin. These antibiotics demonstrated high activity against the pathogen in multiple patients, highlighting their effectiveness as treatment options. Conversely, the least efficient antibiotics were rifampicin, ciprofloxacin, ofloxacin, cefepime, and tigecycline, which exhibited resistance in some cases, indicating limited effectiveness in treating *P. aeruginosa* infections.

Based on data from nine patients with *S. pneumoniae* infections, several antibiotics demonstrated high efficacy across multiple cases. Cefepime, cefotaxime, ceftriaxone, linezolid, and vancomycin were highly active in all nine patients, making them top choices for treatment. Clindamycin, chloramphenicol, erythromycin, rifampicin, imipenem, meropenem, and moxifloxacin were also effective in multiple cases. However, resistance was observed with penicillin in five patients and clarithromycin in several others. Biseptol showed resistance in seven patients, while levofloxacin and ofloxacin were resistant in four patients each. Doxycycline and tetracycline showed resistance in five patients each, highlighting their limited efficacy. Linezolid remained highly effective in seven patients but showed resistance in two, while vancomycin was effective in eight out of nine patients, with one showing resistance. These results underscore the complexity of treating *S. pneumoniae* infections, where antibiotic effectiveness varies and careful selection based on susceptibility is crucial.

For the seven patients with *E. coli* infections, ciprofloxacin, meropenem, and imipenem consistently showed high activity. Gentamicin and amikacin were effective in five patients each, and ceftazidime/ceftazidime + avibactam, along with aztreonam, also showed high activity in multiple cases, making them reliable treatment options.

Several antibiotics were found to be less effective, with high resistance rates in multiple cases. Ampicillin showed resistance in four patients, indicating that it is often ineffective against *E. coli*. Unasyn (ampicillin/sulbactam) and Augmentin (amoxicillin/clavulanate) were resistant in three and two patients, respectively, highlighting reduced efficacy. Cefazolin, tetracycline, and chloramphenicol exhibited resistance in two patients each. Tobramycin, while effective in some cases, showed resistance in one patient, indicating that it may not always be reliable. Cefepime displayed high activity in some patients but resistance in others, and Tazocin (piperacillin/tazobactam) also showed varying results. This variability emphasizes the need for careful antibiotic selection based on individual susceptibility. Overall, ciprofloxacin, meropenem, imipenem, gentamicin, amikacin, ceftazidime/ceftazidime + avibactam, and aztreonam were the most efficient antibiotics for *E. coli*, while ampicillin, Unasyn, Augmentin, cefazolin, tetracycline, chloramphenicol, and tobramycin were less effective, highlighting the importance of tailored antibiotic therapy. An examination of the data from the seven patients infected with methicillin-sensitive *Staphylococcus aureus* (MSSA) highlights several antibiotics demonstrating notable activity against the pathogen, alongside instances of resistance, illuminating the complexities in treating MSSA infections. The antimicrobial susceptibility testing conducted on MSSA revealed varying levels of effectiveness and resistance among the antibiotics tested. Among the most effective antibiotics were amikacin, ciprofloxacin, clarithromycin, clindamycin, chloramphenicol, erythromycin, gentamicin, linezolid, moxifloxacin, and oxacillin, each demonstrating substantial activity against MSSA across multiple cases. Conversely, several antibiotics encountered significant resistance, including ampicillin, piperacillin, penicillin, and Biseptol, highlighting their limited efficacy in combating MSSA infections. These findings underscore the importance of tailored antibiotic selection for MSSA infections.

For instance, amikacin demonstrated high activity in patient 2, indicating its efficacy in this case. However, resistance was observed in patients 6 and 8, underscoring the variability in antibiotic effectiveness against MSSA. Similarly, linezolid exhibited high activity in patients 2, 6, and 8, suggesting its effectiveness as a treatment option. Nonetheless, resistance to linezolid was observed in patient 25, emphasizing the challenges in managing MSSA infections and the need for tailored treatment strategies.

Conversely, certain antibiotics encountered resistance in multiple cases, indicating limitations in their efficacy against MSSA. Ampicillin, piperacillin, and penicillin showed resistance in patients 2 and 6, highlighting their reduced effectiveness against MSSA infections. These findings underscore the complexities involved in treating MSSA infections, where antibiotic effectiveness varies between patients. The observed resistance further emphasizes the importance of judicious antibiotic selection based on individual susceptibility profiles to optimize treatment outcomes.

The antimicrobial susceptibility testing for *A. baumannii* across five patients revealed diverse responses to various antibiotics, reflecting the complexity of managing infections caused by this pathogen. Notably, certain antibiotics exhibited robust activity in some patients while encountering resistance in others, illustrating the variability in treatment efficacy. Among the antibiotics showing notable effectiveness were Unasyn (three patients), ceftazidime + avibactam (two patients), tigecycline (one patient), tobramycin (two patients), colistin (four patients), and Biseptol (four patients), which demonstrated substantial activity against *A. baumannii* in multiple cases. Conversely, several antibiotics encountered significant resistance, including doxycycline (one patient), ciprofloxacin (one patient), cefepime (two patients), ceftazidime (two patients), ceftriaxone (one patient), gentamicin (one patient), levofloxacin (one patient), meropenem (one patient), and tazocin (one patient), highlighting their limited efficacy against this pathogen.

The antimicrobial susceptibility testing for *Acinetobacter* species, based on data from three patients, revealed varied responses to different antibiotics, highlighting the diverse nature of susceptibility patterns among these strains. Notably, certain antibiotics demonstrated significant activity in some patients while encountering resistance in others, underscoring the need for tailored treatment strategies. Among the antibiotics displaying notable efficacy were colistin (three patients), Biseptol (two patients), and tigecycline (one patient). Cefotaxime (two patients), ceftriaxone (one patient), imipenem (one patient), meropenem (two patients), tazocin (two patients), tobramycin (one patient), ciprofloxacin (one patient), gentamicin (one patient), levofloxacin (two patients), and tetracycline (two patients) showed limited efficacy against this pathogen.

Antimicrobial susceptibility testing for *Klebsiella* infections, based on data from ten patients, revealed varying levels of efficacy among several antibiotics. Among the most effective were imipenem (five patients), aztreonam (four patients), cefotaxime (four patients), and cefepime (four patients), demonstrating significant activity and making them promising treatment options. Levofloxacin, gentamicin, and meropenem also showed considerable efficacy in three patients each. However, some antibiotics faced notable resistance, with Augmentin (five patients), ampicillin (four patients), and cefazolin (three patients) showing resistance, presenting challenges in treatment. Additionally, ciprofloxacin (three patients), ertapenem (four patients), and tazocin (four patients) encountered resistance, suggesting reduced effectiveness in certain cases of *Klebsiella* infections.

Several antibiotics demonstrated notable efficacy against *Enterococcus* species, including ciprofloxacin, doxycycline, gentamicin, linezolid, nitrofurantoin, tetracycline, tigecycline, and vancomycin, all showing effectiveness in three patients. However, resistance was observed with antibiotics such as penicillin, colistin, Biseptol, miconazole, fluconazole, tobramycin, amikacin, ceftriaxone, aztreonam, cefepime, ceftazidime, ciprofloxacin, imipenem, meropenem, ofloxacin, piperacillin, ticarcillin, and others. Notably, levofloxacin, moxifloxacin, rifampicin, clarithromycin, clindamycin, erythromycin, and minocycline exhibited resistance in all patients, indicating limited treatment options for *Enterococcus* infections.

In the case of *Staphylococcus epidermidis* (*S. epidermidis*) infections, several antibiotics exhibited consistent patterns across all three patients, either in terms of efficacy or resistance. Among those with high activity in all three cases were ciprofloxacin, clindamycin, gentamicin, levofloxacin, and tigecycline, showing their potential as reliable treatment options against *S. epidermidis*. However, tetracycline and vancomycin were consistently resistant across all three patients, underscoring the limited efficacy of these antibiotics in this type of infection.

For the single case of *Streptococcus* species infection, various antibiotics were tested to determine their activity. Among these, rifampicin, tetracycline, tigecycline, nitrofurantoin, vancomycin, cefoxitin, ciprofloxacin, clindamycin, clarithromycin, chloramphenicol, erythromycin, gentamicin, and moxifloxacin all showed high activity in this particular case. However, significant resistance was found with amikacin, trimethoprim/sulfamethoxazole, aztreonam, and tobramycin, highlighting challenges in treating this type of infection.

In the instance of *Elizabethkingia meningoseptica* infection, only one case was observed. Notably, trimethoprim/sulfamethoxazole, levofloxacin, and minocycline demonstrated high activity against the infection, making them promising for treatment. Conversely, the bacterium exhibited resistance to ceftazidime + avibactam, tigecycline, tobramycin, cefepime, cefotaxime, ceftazidime, meropenem, tazocin, tetracycline, chloramphenicol, amikacin, aztreonam, imipenem, and ticarcillin. Piperacillin and ciprofloxacin, though administered, were ineffective against the infection.

In the treatment of a *Proteus* species infection in a single patient, ertapenem and tazocin demonstrated high activity, while Unasyn showed intermediate efficacy. Resistance was noted against tetracycline, ampicillin, Augmentin, cefazolin, ceftazidime + avibactam, cefuroxime, and gentamicin.

For *Staphylococcus haemolyticus*, linezolid, tigecycline, nitrofurantoin, vancomycin, and quinupristin/dalfopristin showed high effectiveness. However, resistance was encountered with tetracycline, penicillin, Biseptol, ciprofloxacin, gentamicin, clindamycin, levofloxacin, and erythromycin.

In the case of *Salmonella* infection, antibiotics such as cefazolin, cefepime, cefoxitin, ciprofloxacin, colistin, fosfomycin, and imipenem were highly effective, with additional activity seen from amikacin, aztreonam, gentamicin, nitrofurantoin, and norfloxacin. Resistance was observed against ampicillin, Biseptol, and tobramycin, indicating limitations for these antibiotics.

In the treatment of *Candida* species, a variety of antifungal agents were utilized. Among these, penicillin, amphotericin B, econazole, flucytosine, fluconazole, itraconazole, and miconazole were administered and had high activity. However, despite the effectiveness of most antifungal agents, voriconazole was found to be resistant to the infection, highlighting the necessity of understanding specific resistance patterns in fungal infections for proper treatment.

## 3. Discussion

Other research groups have reported similar findings regarding the efficacy of certain antibiotics and the resistance patterns of common pathogens. For instance, a study by David et al. on pediatric infections in China found that linezolid and ertapenem were effective against multidrug-resistant strains, corroborating our study’s findings [15]. Another research group, led by Jones et al., demonstrated a significant prevalence of MRSA in pediatric populations, with linezolid and teicoplanin being among the most effective treatments [16]. This aligns with the findings of our study, which also reported the high efficacy of these antibiotics against MRSA [17].

Reynolds et al. observed that carbapenems like ertapenem were highly effective against Gram-negative bacteria such as *P. aeruginosa* and *Klebsiella* spp., which was also prevalent in our study [18]. Similarly, Lin et al. found that lower-generation cephalosporins are increasingly ineffective against common pathogens due to rising resistance levels [19].

The high resistance to penicillin observed in our study is supported by global trends. Gurung et al. indicated widespread penicillin resistance among pediatric bacterial isolates, necessitating the use of more potent antibiotics [20]. In another study, Williams et al. reported that MRSA strains were resistant to many antibiotics, with linezolid and teicoplanin being among the few effective options [21]. This pattern of antibiotic resistance underscores the need for tailored treatment strategies based on susceptibility profiles.

Furthermore, Kim et al. highlighted the critical role of carbapenems and linezolid in treating severe infections like sepsis in pediatric populations, which is consistent with our study’s findings [22]. The variability in cephalosporin resistance observed in our study is consistent with findings from multiple studies, including those by Nguyen et al. and Brown et al., who reported that resistance levels tend to be higher for lower-generation cephalosporins compared to higher-generation ones [23,24].

Other researchers, such as Lee et al., have documented high resistance rates to antibiotics like oxacillin and ampicillin in various bacterial isolates from pediatric patients, mirroring the findings of our study [25]. Additionally, Garcia et al. and Lyu et al. further support these observations, showing that antibiotics such as linezolid and ertapenem remain effective against many resistant strains, while traditional antibiotics like penicillin and ampicillin are largely ineffective due to widespread resistance [26,27].

A study by Su et al. demonstrated a dynamic change in the serotype distribution and antimicrobial resistance of pneumococcal isolates, highlighting the challenges in treating infections with high resistance levels [28]. Similarly, Wu et al. described trends in the antibiotic susceptibility and clonal distribution of *Staphylococcus aureus* in pediatric skin and soft tissue infections over a decade, emphasizing the need for effective antibiotics like linezolid and teicoplanin [29].

Wang et al. presented an antimicrobial resistance profile of MRSA isolates in children, aligning with our study’s findings on the efficacy of linezolid and teicoplanin [30]. Markovska et al. characterized resistance genes and plasmids from sick children with *Salmonella enterica*, underscoring the complexity of treating such infections with rising resistance levels [31].

Xiao et al. described the fecal carriage rate of extended-spectrum β-lactamase-producing or carbapenem-resistant Enterobacterales among Japanese infants, reflecting the global challenge of antibiotic resistance [32]. Chen et al. presented the antibiotic susceptibility of *Escherichia coli* isolated from neonates, showing high resistance to commonly used antibiotics [33].

Weidmann et al. discussed the clinical significance of macrolide resistance in pediatric *Mycoplasma pneumoniae* infection, further highlighting the necessity for alternative treatments like those identified in our study [34]. Nahata et al. assessed respiratory viral exclusion and affinity interactions through coinfection incidence in pediatric populations, providing additional context to the complexity of treating antibiotic-resistant infections [35].

Romandini et al. demonstrated that vancomycin dosages of 40–60 mg/kg/day are effective, paralleling our study’s findings on the efficacy of linezolid [36]. Pani et al. discussed global emerging threats of antibiotic resistance in pediatric infections, emphasizing the need for continuous surveillance and tailored antibiotic therapies [37].

Schenardi et al. highlighted the challenges in treating pediatric infections with high resistance rates, supporting our study’s call for more effective antibiotics like linezolid and ertapenem [38]. Other researchers have echoed these sentiments, underscoring the need for judicious antibiotic use to combat rising resistance levels [39]. Tian et al. showed that specific antibiotics remain effective against resistant bacteria when used appropriately, reinforcing our study’s findings [40]. Muteeb et al. emphasized the importance of strategic antibiotic use and ongoing research to develop new treatments, aligning with the recommendations of our study [41].

Our study revealed that antibiotics such as linezolid and ertapenem remain highly effective against multidrug-resistant organisms, similar to findings from studies conducted in other regions, including China and Europe. These studies, like ours, have consistently shown that antibiotics such as linezolid and teicoplanin are potent against MRSA, while carbapenems like ertapenem are effective against Gram-negative bacteria. While it is well known that Romania has one of the highest rates of antibiotic consumption in Europe, which could contribute to higher levels of resistance, our findings did not show an overwhelming increase in resistance compared to other countries. This could be due to recent efforts to control antibiotic misuse, including public health campaigns and stricter regulations on antibiotic sales without prescriptions. As noted, resistance to common antibiotics like penicillin and oxacillin was indeed high, which aligns with both national and global trends. However, the efficacy of newer-generation antibiotics suggests that the targeted use of potent antibiotics is helping to manage resistance levels effectively. We observed variability in resistance patterns across different classes of antibiotics. Lower-generation cephalosporins exhibited higher resistance, consistent with studies from other regions. This suggests that resistance may not solely be driven by overuse, but also by the specific types of antibiotics being used more frequently in certain contexts. For instance, the widespread use of lower-generation antibiotics often leads to higher resistance in those categories, whereas a more judicious use of higher-generation antibiotics can preserve their efficacy.

## 4. Materials and Methods

The study group included 31 children admitted to the Pediatric Clinic II/Oncopediatrics Department of the County Emergency Clinical Hospital of Craiova from January 2020 to January 2024, with various conditions (pneumonia, bronchopneumonia, urinary infections, septicemia, severe neurological conditions, acute lymphoblastic leukemia). Cultures were collected from these children (nasal swab, pharyngeal swab, tracheobronchial aspirate, blood culture, gastric lavage fluid culture, conjunctival secretion culture, skin culture, central venous catheter culture). The ages of the children ranged from newborns to 13 years old. The gender distribution was 23 males and 8 females. Based on the environment of origin, there were 9 children from urban areas and 21 from rural areas (including 2 institutionalized children). To test the antibacterial effect, we used the Kirby–Bauer disk diffusion test, according to FR X [42,43,44,45]. All children received antibiotic treatment based on antibiogram results. The treatment duration varied between 7 and 10 days, depending on the specific pathology. Additionally, the antibiotic dosages were calculated based on the patient’s body weight (mg/kg), ensuring personalized and appropriate treatment for each individual.

### Tested Antibiotics

Beta-lactams:
❖Penicillins:AmpicillinPenicillinPiperacillinOxacillinUnasyn (ampicillin/sulbactam)TicarcillinAugmentin (amoxicillin/clavulanate)Tazocin (piperacillin/tazobactam)❖Cephalosporins:Cefepime (4th generation)Cefotaxime (3rd generation)Ceftriaxone (3rd generation)Cefoxitin (2nd generation)Ceftaroline (5th generation)Ceftazidime (3rd generation)Ceftazidime + avibactam (3rd generation with beta-lactamase inhibitor)Cefotaxime + avibactamCefuroxime (2nd generation)Cefazolin (1st generation)Cefort (cefotetan, 2nd generation)❖Carbapenems:ImipenemMeropenemErtapenem❖Monobactams:Aztreonam
Macrolides:ClarithromycinErythromycinTetracyclines:DoxycyclineTetracyclineMinocyclineTigecyclineAminoglycosides:AmikacinGentamicinTobramycinFluoroquinolones:Ciprofloxacin (Ciprinol, Ciproxina)LevofloxacinOfloxacinMoxifloxacin (Avelox)Sulfonamides:Biseptol (trimethoprim/sulfamethoxazole)Oxazolidinones:LinezolidLincosamides:ClindamycinPolypeptides:ColistinRifamycins:RifampicinPhenicol:ChloramphenicolPhosphonate:Fosfomycin (Monural)Nitrofurans:NitrofurantoinLipopeptide:DaptomycinOthers:Vancomycin (glycopeptide)Teicoplanin (glycopeptide)Sulcef (sultamicillin)Chlorhexidine

## 5. Conclusions

Our study provides a comprehensive analysis of antibiotic resistance patterns among pediatric patients in Romania, highlighting the prevalence of multidrug-resistant organisms (MDROs) and the effectiveness of various antibiotics. The findings underscore significant resistance to commonly used antibiotics like penicillin, oxacillin, and ampicillin, necessitating the reconsideration of these drugs as first-line treatments. Linezolid, ertapenem, and teicoplanin emerged as the most effective antibiotics, offering promising options for managing resistant infections. The study emphasizes the critical need for tailored antibiotic therapy based on susceptibility profiles to improve treatment outcomes and combat antibiotic resistance effectively. Additionally, it highlights the importance of ongoing surveillance and strategic antibiotic use to mitigate the rising threat of AMR in Romania, where the impact is particularly severe. This research contributes valuable insights into local resistance patterns and effective treatments, informing public health strategies and clinical practices to address the challenge of antibiotic resistance in pediatric infections.

## Figures and Tables

**Figure 1 antibiotics-13-00879-f001:**
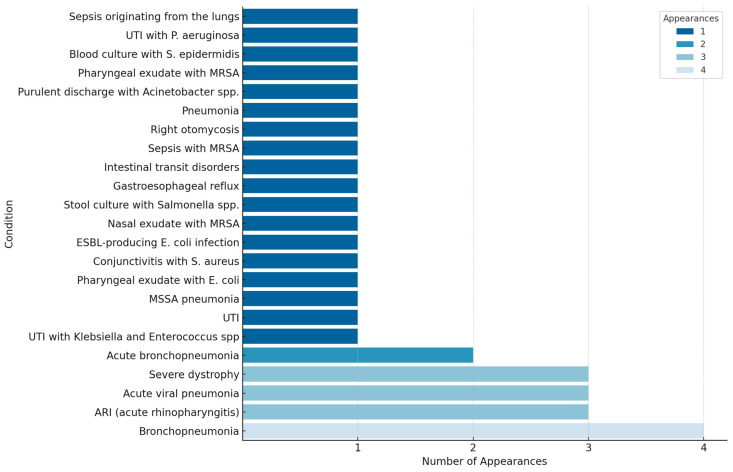
Frequency distribution of pediatric conditions by number of appearances.

**Figure 2 antibiotics-13-00879-f002:**
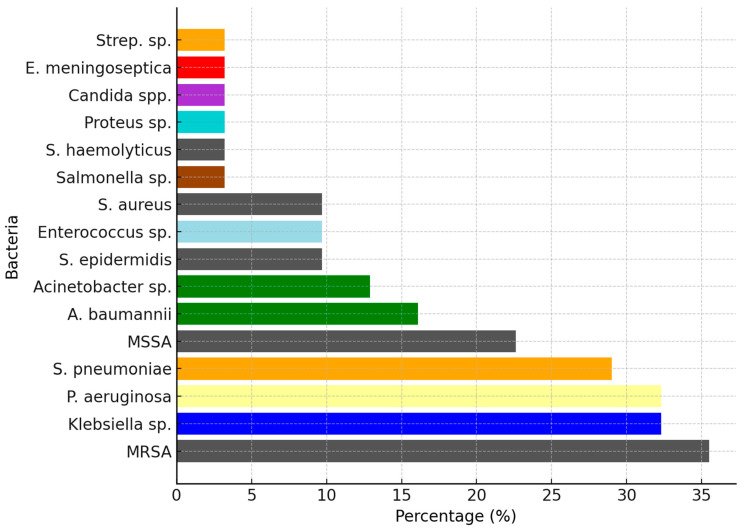
Percentages of different bacterial pathogens identified in the study population, representing the global proportion of each bacterial type across all pathologies. Bacterial species have been highlighted with distinct colors (*Staphylococcus* species: gray; *Acinetobacter* species: green; *Streptococcus* species: orange).

**Figure 3 antibiotics-13-00879-f003:**
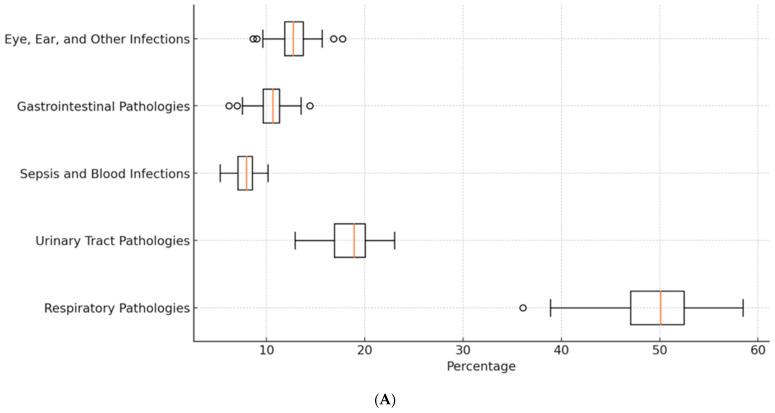
(**A**) Percentages of different pathologies diagnosed in our patients. (**B**) Age distribution of pathologies in pediatric patients.

**Figure 4 antibiotics-13-00879-f004:**
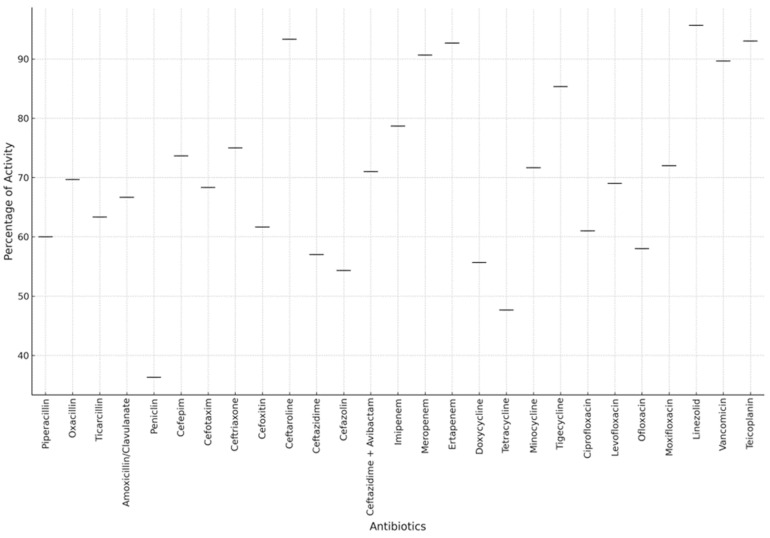
Global antibiotic activity against bacterial pathogens in pediatric patients.

**Table 1 antibiotics-13-00879-t001:** Patient demographics, diagnoses, and identified bacterial pathogens.

Age	Diagnosis	Pathogens
2 M	Bronchopneumonia, ARI	*E. coli*, *S. pneumoniae*, *P. aeruginosa*
12 D	Bronchopneumonia, ARI, acute rhinopharyngitis	*MSSA*, *S. pneumoniae*, *Streptococcus* sp.
1 Y	Acute viral pneumonia, ARI, severe dystrophy	*A. baumanii*, *MRSA*, *P. aeruginosa*, *E. meningoseptica*
2 M	Acute bronchopneumonia, ARI	*E. coli*, *S. pneumoniae*
3 M	UTI, acute diarrheal illness	*Klebisiella*, *MRSA*
5 M	MSSA pneumonia, pharyngeal exudate with *E. coli*	*MSSA*, *E. coli*
12 D	Pneumonia, conjunctivitis with *S. aureus*	*Acinetobacter* spp., *S. aureus*
2 M	Acute bronchopneumonia	*S. pneumoniae*, *MSSA*
6 M	ESBL-producing *E. coli* infection	*E. coli*
8 D	UTI with *Klebsiella* and *Enterococcus* spp.	*Klebsiella*, *Enterococcus* spp.
4 M	Sepsis originating from the lungs, bronchopneumonia, ARI	*Enteroccocus* spp., *A. baumanni*, *S. aureus*
8 M	Gastroesophageal reflux, intestinal transit disorders	*Klebsiella* spp., *E. coli*, *S. pneumoniae*, *Candida* spp.
1 Y	Bronchopneumonia, ARI, sepsis with *MRSA*	*S. pneumoniae*, *P. aeruginosa*, *MRSA*
9 M	Gastroesophageal reflux, right otomycosis	*S. pneumoniae*, *P. aeruginosa*, *A. baumanni*
11 M	Bullous impetigo, acute conjunctivitis, diarrhea	*Enteroccocus* spp., *Klebsiella* spp., *S. aureus*
18 M	Acute pneumonia, acute rhinopharyngitis	*MRSA*, *Staphilococcus epiderdimitis*, *Staphilococcus haemoliticus*, *P. aeruginosa*
10 M	Acute pneumonia, diarrhea, Ebstein’s anomaly	*MRSA*, *P. aeruginosa*
4 Y	Sepsis with *MRSA*, anemia, secondary dyspepsia, UTI	*MRSA*, *Acinetobacter* spp., *Klebsiella*
3 Y	Acute pneumonia, unspecified sepsis	*Preoteus* spp., *Klebsiella*
2 Y	Sepsis with *S. aureus*, bronchopneumonia	*S. pneumonie*, *MRSA*, *P. aeruginosa*
5 Y	Acute lymphoblastic leukemia, conjunctivitis, sepsis	*Acinetobacter baumanii*, *Acinetobacter* spp.
6 W	Pharyngitis with *MSSA*	*MSSA*, *E. coli*, *P. aeruginosa*
5 Y	Pneumonia	*MRSA*, *S. pneumonie*
13 Y	Purulent discharge with *Acinetobacter* spp.	*Acinetobactr* spp.
4 Y	Pharyngeal exudate with *MRSA*	*MRSA*, *Staphylococcus epiderdimitis*
5 Y	Blood culture with *S. epidermidis*	*MSSA*, *Staphylococcus epiderdimitis*
6 M	UTI with *P. aeruginosa*	*Klebsiella* spp., *P. aeruginosa*
1 M	Stool culture with *Salmonella* spp., pharyngeal exudate with *MRSA*, nasal exudate with *MRSA*	*Salmonella* spp., *MRSA*, *MSSA*
18 M	Bronchopneumonia, ARI	*E. coli*, *P. Aerugonisa*, *MSSA*, *Klebsiella* spp.
2 Y	Acute pneumonia, UTI	*Klebsiella*, *MRSA*
6 M	Acute pneumonia	*Acinetobacter baumanii*, *Klebsiella*

**Table 2 antibiotics-13-00879-t002:** Age, diagnosis, detected pathogens, and the effectiveness of various antibiotic classes.

Patient nr.	Age	Diagnosis	Pathogens	PN	C	CB	M	T	A	F	S	O	P	N	G
1.	2M	Bronchopneumonia, ARI	*E. coli*, *S. pneumoniae*, *P. aeruginosa*	H	H	H	H	R	H	H	R	H	H	/	H
2.	12 D	Bronchopneumonia, ARI, Acute Rhinopharyngitis	MSSA, *S. pneumoniae*, *Streptococcus* sp.	H	H	H	H	R	I	I	R	**H**	/	/	H
3.	1 Y	Acute Viral Pneumonia, ARI, Severe Dystrophy	*Acinetobacter baumanii*, *MRSA*, *P. aeruginosa*, *Elizabethkingia meningoseptica* *	I	I	H	H	I	H	**H**	**H**	H	H	/	/
4.	2 M	Acute bronchopneumonia, ARI	*E. coli*, *S. pneumoniae*	I	H	H	H	H	H	H	R	H	H	H	H
5.	3 M	UTI, acute diarrheal illness	*Klebisiella*, MRSA	R	R	I	R	H	I	H	R	H	/	R	/
6.	5 M	MSSA pneumonia, Pharyngeal exudate with *E. coli*	MSSA, *E. coli*	H	H	H	R	H	H	/	I	H	/	H	/
7.	12 D	Pneumonia, Conjunctivitis with *S. aureus*	*Acinetobacter* spp., *S. aureus*	I	R	R	H	H	H	H	H	/	H	/	/
8.	2M	Acute bronchopneumonia	*S. pneumoniae*, MSSA	H	H	H	I	H	H	R	R	**H**	/	/	**H**
9.	6 M	ESBL-producing *E. coli* infection	*E. coli*	R	R	H	/	H	R	/	H	/	H	/	/
10.	8 D	UTI with *Klebsiella* and *Enterococcus* spp.	*Klebsiella* spp., *Enterococcus* spp.	R	H	H	/	H	H	I	H	H	H	R	H
11.	4 M	Sepsis originating from the lungs, bronchopneumonia, ARI	*Enteroccocus* spp., *Acinetobacter baumanni*, *S. aureus*	R	R	R	R	I	I	I	H	H	I	I	/
12.	8M	Gastroesophageal reflux, intestinal transit disorders.	*Klebsiella* spp., *E. coli*, *S. pneumoniae*, *Candida* spp. *	H	H	H	H	R	H	H	H	H	/	H	H
13.	1Y	Bronchopneumonia, ARI, sepsis with MRSA	*S. pneumoniae*, *P. aeruginosa*, MRSA	I	/	R	R	R	I	H	I	**H**	/	/	**H**
14.	9 M	Gastroesophageal reflux, right otomycosis	S. *pneumoniae*, *P. aeruginosa*, *Acinetobacter baumanni*	I	H	I	R	R	I	I	R	H	H	/	H
15.	11 M	Bullous impetigo, acute conjunctivitis, diarrhea	*Enteroccocus* spp., *Klebsiella* spp., *S. aureus*	H	H	H	H	I	H	H	H	**H**	H	/	/
16.	10 M	Acute pneumonia, diarrhea, Ebstein’s anomaly	MRSA, *P. aeruginosa*	I	H	R	R	R	I	**H**	R	**H**	H	/	/
17.	4Y	Sepsis with MRSA, anemia, secondary dyspepsia, UTI	MRSA, *Acinetobacter* spp., *Klebsiella* spp.	R	I	I	R	R	R	R	I	H	H	/	H
18.	3Y	Acute pneumonia, Unspecified sepsis	*Preoteus* spp., *Klebsiella*	I	I	**H**	/	I	R	/	H	/	I	/	/
19.	2Y	Sepsis with *Staphylococcus aureus*, bronchopneumonia	*S. pneumonie*, *MRSA*, *P. aeruginosa*	I	/	H	R	R	I	**H**	/	/	/	/	H
20.	5Y	Acute lymphoblastic leukemia, conjunctivitis, sepsis	*Acinetobacter baumanii*, *Acinetobacter* spp.	I	R	R	/	R	I	/	H	/	H	/	/
21.	6 W	Pharyngitis with MSSA	MSSA, *E. coli*, *P. aeruginosa*	I	I	H	R	H	H	H	H	H	H	/	/
22.	5 Y	Pnemunonia	MRSA, *S. pneumonie*	I	H	H	I	H	I	H	I	H	/	/	H
23.	13 Y	Purulent discharge with *Acinetobacter* spp.	*Acinetobacter* spp.	I	I	R	/	/	I	/	H	/	H	/	/
24.	4 Y	Pharyngeal exudate with MRSA	MRSA, *Staphylococcus epiderdimitis*	I	/	/	R	/	I	H	H	I	/	/	H
25.	5 Y	Blood culture with S. epidermidis	MSSA, *Staphylococcus epiderdimitis*	I	/	/	H	H	H	H	H	H	/	H	H
26.	6 M	UTI with *P. aeruginosa*	*Klebsiella* spp., *P. aeruginosa*	I	R	H	/	R	H	I	R	/	/	/	/
27.	1 M	Stool culture with *Salmonella* spp., Pharyngeal exudate with MRSA, Nasal exudate with MRSA	*Salmonella* spp., MRSA, MSSA	I	H	H	H	H	H	H	I	H	H	/	H
28.	18M	Bronchopneumonia, ARI	*E. coli*, *P. Aerugonisa*, *MSSA*, *Klebsiella* spp.	I	H	I	H	H	H	H	H	H	**H**	/	/
29.	2Y	Acute pneumonia, UTI	*Klebsiella* spp., MRSA	R	H	H	R	I	H	/	H	H	/	H	/
30.	6 M	Acute pneumonia	*Acinetobacter baumanii*, *Klebsiella* spp.	I	H	I	/	H	I	I	I	/	H	/	/
31.	18 M	Acute pneumonia, Acute Rhinopharyngitis	MRSA, *Staph. Epiderdimitis*, *Staph. Haemoliticus*, *P. aeruginosa*	I	H	H	I	I	H	H	I	H	I	/	H

I = intermediate, H = high activity, **H** = active on all bacteria that patient had, R = resistant, / = not tested; * acute respiratory infection, PN = beta-lactams, C = cephalosporins, CB = carbapenems, M = macrolides, T = tetracyclines, A = aminoglycosides, F = fluoroquinolones, S = sulfonamides, O = oxazolidinones, P = polypeptides, N = nitrofurans, G = glycopeptides.

## Data Availability

Data are contained within the article.

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
