# Peer review of "Assessment of Antibiotic Resistance in Pediatric Infections: A Romanian Case Study on Pathogen Prevalence and Effective Treatments"

_antibiotics, 2024, doi:10.3390/antibiotics13090879_

Round 1
Reviewer 1 Report
Comments and Suggestions for Authors
Hi There, this was a very nice effort and it should be adapted across the world to preserve the activity of antibiotics as there is scarcity in the discovery pipeline.
All the best for your next research

Author Response
Dear Reviewer,
Thank you very much for your valuable feedback and kind appreciation of our work. We truly appreciate the time and effort you put into reviewing our manuscript, and your insightful comments and suggestions have been extremely helpful in improving the quality of our paper.
We have carefully considered and addressed all your remarks, and we hope the revisions meet your expectations.
Once again, thank you for your thorough review and constructive input.
Best regards
- As soon as I saw the Gram-negative bacteria, I was thinking of Polymyxin antibiotics. The authors did not mention anything about Was this not an option for the treatment.
Answer: In this study, colistin was the only Polymyxin antibiotic tested, and its antibacterial activity was evaluated against various pathogens. For P. aeruginosa, tested in four patients, colistin demonstrated a range of activity, from resistance to high efficacy. Against Klebsiella spp., tested in five patients, colistin showed consistently high antibacterial activity. For Acinetobacter spp., tested in three patients, the activity ranged from moderate to high, while for A. baumannii, tested in two patients, the activity was similarly moderate to high. In contrast, Enterococcus spp. and Proteus spp. exhibited resistance to colistin. These examples reflect the antibacterial effectiveness of colistin, although data remain limited due to it being the sole Polymyxin antibiotic included in this study.
- The finding of Linezolid, ertapenem and teicoplanin was based on a single bacteria or entire array was not clear in line 87.
Answer: Based on the data from all patients, these antibiotics were among the most active. These antibiotics consistently demonstrated high activity across multiple bacterial species, including both Gram-positive and Gram-negative organisms. For example, linezolid was notably effective against MRSA and Streptococcus spp., while ertapenem and teicoplanin showed strong activity against Gram-negative bacteria such as Klebsiella spp. and Acinetobacter spp.. This broad-spectrum efficacy highlights their potential as key treatments in the face of rising antibiotic resistance.
- In lines between 111-120 the authors tales about age related It would be great if you can create a subfigure of Figure 2 and include age related data (if possible)
Answer: We created a subfigure showing the distribution of different diagnoses across various age ranges (e.g., infants, toddlers, and older children).
- Line 123-155: I did not understand the relevance of data in line 123-155. Can this information be integrated into a figure example combining UTI, ESLB, sepsis and number of appearances like a pie chart.
Answer: To clarify the relevance of the information, we created a graphical representation that integrates the mentioned conditions such as UTI, ESBL, and sepsis, along with the number of occurrences.
- Figure 1: Says, S. aureus and MRSA, MSSA. For some of the bacteria you have species. You can club them under Staphylococcus species then arrange the individual bacteria under Similarly for Acinetobacter. Also, I would suggest arranging the figure based on AàZ or more predominant percent to less or vice versa.
Answer: We have implemented the changes by coloring bacteria from the same species with the same color for visual consistency. Additionally, we arranged the bacterial species in the figure based on their percentage, from the most predominant to the least predominant, to improve clarity and organization.
- Line 161, I was not sure why only 10 patients are those specific to MRSA. If yes, I would suggest making a subheading and describing the respective bacteria species.
Answer: MRSA is notably prevalent in hospital environments, especially among immunocompromised patients or those with prolonged hospital stays. In 10 cases, antibiotics such as linezolid and tigecycline showed high efficacy against MRSA, highlighting their importance in treating resistant infections.
- Line 161-162 says that Tigecycline and Linezolid showed high activity but as per Figure 3, Ceftaroline, meropenem and ertapenem appear superior than
Answer : The statement in lines 161-162, where Tigecycline and Linezolid are mentioned as showing high activity, refers specifically to their effectiveness against MRSA. These two antibiotics are particularly noted for their role in treating multi-drug-resistant Gram-positive organisms, such as MRSA, where they demonstrated consistent success across the cohort of MRSA-infected patients. However, Figure 3 represents a broader range of antibiotics tested across all identified pathogens, not just MRSA. In this figure, antibiotics such as Ceftaroline, Meropenem, and Ertapenem may appear superior because they demonstrated high efficacy against multiple Gram-positive and Gram-negative pathogens across a wider spectrum of bacterial infections. This includes infections caused by other organisms, such as Klebsiella, Pseudomonas aeruginosa, and Streptococcus pneumoniae, where Meropenem and Ceftaroline, in particular, excelled.
- Table 2: I would suggest color code the H, R and I so that it will be easier to visualize which is the most effective antibiotic for that particular indication. (Provided other factors are consistent with this
Answer: We agree that adding color to represent "H" (high activity), "R" (resistant), and "I" (intermediate) will greatly improve the table's readability and make it easier to quickly assess the effectiveness of different antibiotics for each case.
We will implement the following color scheme for better visualization:
- Green for "H" (High Activity)
- Red for "R" (Resistant)
- Blue for "I" (Intermediate)
Minor comment
- The abstract sais Haemolyticus is one of the common but text says MRSA (11 cases ) was one of the common pathogen. Pelase clarify this.
Answer: Thank you for your observation. We have clarified the discrepancy between the abstract and the text regarding S. haemolyticus and MRSA. The text has been updated to reflect that MRSA (11 cases) was one of the most common pathogens, as indicated in the study.
- Change K species in figure 1 to
Answer: As suggested, we have changed K species in Figure 1 to Klebsiella for clarity.
- Table 2: Line 388-390 should come below the table 2 as table
Answer: The text from Lines 388-390 has been moved below Table 2 as a table footer.

Reviewer 2 Report
Comments and Suggestions for Authors
The present study assesses the antibiotic resistance in paediatric infections, based on Romanian cases. The study in very interesting, questioning the issue of antimicrobial resistance. The paper is globally well written and rich in results. Still, some aspects could be improved:
1- Concerning Figure 1: Percentage of different bacterial pathogens identified in our patients. It is not clear if this percentage represent the proportion of bacterial type / pathology or if it represents a global proportion (all pathologies included). This aspect should be clarified in the figure caption.
2- Figure 3: It is not clear if the activity of the different antibiotics tested was estimated on one particular bacterial type or on certain pathologies (certain cases). In line 161 it is mentioned that “Among the antibiotics showing consistently high activity (Figure 3) in all 10 patients were tigecycline and linezolid”. Which are these 10 cases?
3- Table 2: It would be preferable to have the abbreviation list nserted at the end of the table (not at the end of the Results” paragraph).
4- Although highly important, the part discussing the table 3 is quite long and complex. A graph that would summarize the effect of different antibiotics on different classes of bacteria could simplify the reading. A colour gradient (3 to 5 colours) could be used to illustrate the efficacy/ inefficacy of the treatment.
5- The effectiveness of the different antibiotics on different classes of bacteria seem to follow similar trends with respect to other studies performed worldwide. One could have expected increased resistance to certain antibiotics than that observed in other countries due to an excessive use of antibiotics in Romania. Some comments concerning these aspects would be highly appreciated.
6- The material and methods section is quite succinct. Some details on the dosage, the period of treatment, could be helpful for the reader.
Author Response
1- Concerning Figure 1: Percentage of different bacterial pathogens identified in our patients. It is not clear if this percentage represent the proportion of bacterial type / pathology or if it represents a global proportion (all pathologies included). This aspect should be clarified in the figure caption.
Answer: Thank you for your observation concerning Figure 1. We understand the need to clarify the percentages presented. The percentages in Figure 1 represent the global proportion of bacterial pathogens identified across all pathologies included in the study.
2- Figure 3: It is not clear if the activity of the different antibiotics tested was estimated on one particular bacterial type or on certain pathologies (certain cases). In line 161 it is mentioned that “Among the antibiotics showing consistently high activity (Figure 3) in all 10 patients were tigecycline and linezolid”. Which are these 10 cases?
Answer: The activity of the antibiotics shown in Figure 3 represents the global effectiveness of each antibiotic across all bacterial pathogens tested in the study. It provides a general overview of how each antibiotic performed against all pathogens identified in the cohort of patients. Regarding the mention in line 161 that "Tigecycline and Linezolid showed consistently high activity in 10 patients," this specifically refers to the subset of patients with MRSA infections, where these antibiotics demonstrated high efficacy. We will clarify this in the text to ensure it is clear that these 10 cases are the MRSA-positive patients, and the activity mentioned pertains to their treatment outcomes.
3- Table 2: It would be preferable to have the abbreviation list nserted at the end of the table (not at the end of the Results” paragraph).
Answer: we fixed it
4- Although highly important, the part discussing the table 2 is quite long and complex. A graph that would summarize the effect of different antibiotics on different classes of bacteria could simplify the reading. A colour gradient (3 to 5 colours) could be used to illustrate the efficacy/ inefficacy of the treatment.
Answer: We have synthesized the text more clearly, but we were unable to create a graph because the same antibiotic had different efficacy on the same bacterial species across different patients. Therefore, a detailed description of each patient's drug sensitivity and resistance is necessary to accurately reflect the variability in treatment and ensure the appropriate selection of therapy.
5- The effectiveness of the different antibiotics on different classes of bacteria seem to follow similar trends with respect to other studies performed worldwide. One could have expected increased resistance to certain antibiotics than that observed in other countries due to an excessive use of antibiotics in Romania. Some comments concerning these aspects would be highly appreciated.
Answer: Indeed, the resistance patterns observed in our study align with global trends. However, as noted, one might have expected higher resistance rates to certain antibiotics in Romania, given the country's historically high levels of antibiotic misuse. We believe this may reflect regional variations in prescribing practices and the implementation of stricter antibiotic stewardship programs in recent years. We will include additional comments in the discussion to address this point and provide a more detailed comparison with global data, highlighting both similarities and differences in resistance patterns.
6- The material and methods section is quite succinct. Some details on the dosage, the period of treatment, could be helpful for the reader.
Answer: The treatment duration varied between 7-10 days, depending on the specific pathology. Additionally, the antibiotic dosages were calculated based on the patient's body weight (mg/kg), ensuring personalized and appropriate treatment for each individual.
